# Ehlers–Danlos Syndrome and Hypermobility Syndrome Compared with Other Common Chronic Pain Diagnoses—A Study from the Swedish Quality Registry for Pain Rehabilitation

**DOI:** 10.3390/jcm9072143

**Published:** 2020-07-07

**Authors:** Peter Molander, Mehmed Novo, Andrea Hållstam, Monika Löfgren, Britt-Marie Stålnacke, Björn Gerdle

**Affiliations:** 1Pain and Rehabilitation Centre and Department of Health, Medicine and Caring Sciences, Linköping University, 581 83 Linköping, Sweden; bjorn.gerdle@liu.se; 2Department of Behavioural Sciences and Learning, Linköping University, 581 83 Linköping, Sweden; 3Department of Community Medicine and Rehabilitation, Rehabilitation Medicine, Umeå University, SE-905 87 Umeå, Sweden; Mehmed.Novo@regionvasterbotten.se (M.N.); britt-marie.stalnacke@umu.se (B.-M.S.); 4Department of Clinical Sciences, Danderyd Hospital, Karolinska Institute, SE-182 88 Stockholm, Sweden; andrea.hallstam@sll.se (A.H.); monika.lofgren@ki.se (M.L.)

**Keywords:** Ehlers–Danlos syndrome, hypermobility syndrome, pain, chronic pain

## Abstract

Although chronic pain is common in patients with Ehlers–Danlos syndrome (EDS) and hypermobility syndromes (HMS), little is known about the clinical characteristics of these groups. The main aim was to compare EDS/HMS with common local and generalized pain conditions with respect to Patient Reported Outcome Measures (PROMs). Data from the Swedish Quality Register for Chronic Pain (SQRP) from 2007 to 2016 (*n* = 40,518) were used, including patients with EDS/HMS (*n* = 795), fibromyalgia (*n* = 5791), spinal pain (*n* = 6693), and whiplash associated disorders (WAD) (*n* = 1229). No important differences in the PROMs were found between EDS and HMS. Women were represented in > 90% of EDS/HMS cases and fibromyalgia cases, and in about 64% of the other groups. The EDS/HMS group was significantly younger than the others but had a longer pain duration. The pain intensity in EDS/HMS was like those found in spinal pain and WAD; fibromyalgia had the highest pain intensity. Depressive and anxiety symptoms were very similar in the four groups. Vitality—a proxy for fatigue—was low both in EDS/HMS and fibromyalgia. The physical health was lower in EDS/HMS and fibromyalgia than in the two other groups. Patients with EDS/HMS were younger, more often female, and suffered from pain for the longest time compared with patients who had localized/regional pain conditions. Health-care clinicians must be aware of these issues related to EDS/HMS both when assessing the clinical presentations and planning treatment and rehabilitation interventions.

## 1. Introduction

Ehlers–Danlos syndrome (EDS) is a heritable condition characterized by a disorder in the collagen, causing increased laxity in bodily organs such as skin, ligament, joints, blood vessels, and inner organs [1]. As a consequence, EDS patients can have problems such as joint hypermobility, skin hyperextensibility, and atrophic scarring, cardiovascular abnormalities, dysautonomia, chronic fatigue, anxiety, depression [2,3], and structural and functional anomalies in the gastrointestinal canal [4]. EDS includes 13 sub-types [1] with the hypermobility sub-type the most common, with a prevalence of about 1% [5]. Generalized hypermobility in childhood increases the risk of joint pain during teenage years [6]. In addition, joint hypermobility is often associated with daily pain and increases with age, negatively affecting the physical function of young people [6,7]. A prevalence study found that 18% of the general population self-reported joint hypermobility [8]. If joint hypermobility affects many joints, it is classified as hypermobility syndrome (HMS) [9,10]. EDS hypermobility type, the most common sub-type, has similar symptoms and diagnostic criteria as HMS, so clinically discriminating between them can be challenging [9]. Before 2017, the Villefranche nosology was a common tool for diagnosis of 6 different forms of EDS [11] as well as the Brighton criteria for HMS [12]. An overview of the changes is available in Appendix A.

Chronic pain, a major problem both in EDS and HMS, is found in up to 90% of patients with hypermobile EDS [13,14]. The pain is generally complex, often generalized [15], and for people with hypermobile EDS often leads to poor physical, psychosocial, and overall function comparable to patients with fibromyalgia (FM) [16]. As in other chronic pain conditions, psychological symptoms are common both in EDS and HMS [14,17]. In a recent study on EDS where the majority had the hypermobility type, 51% of respondents reported high levels of anxiety and 20% reported high levels of depression [14]. Fatigue, sleep problems, and other subjective health complaints that influence quality of life are also more frequent in hypermobile EDS/HMS than in controls [18,19].

To date, there are few comprehensive studies of the self-reported clinical situation for patients with EDS/HMS and a lack of knowledge of how the clinical picture (e.g., pain aspects, emotional distress, and quality of life) relates to other common chronic pain conditions representing local and generalized pain conditions. This knowledge gap has motivated this study of chronic pain patients based on Patient Reported Outcome Measures (PROMs) from the Swedish Quality Registry for Pain Rehabilitation (SQRP). The SQRP offers an opportunity to investigate PROMs as most relevant specialist care units throughout Sweden deliver data to the SQRP. Hence, this study compares EDS/HMS with common local and generalized chronic pain conditions/diagnoses with respect to PROMs such as pain aspects, emotional distress, life impact, and health-related quality of life (HRQoL) in large cohorts of patients.

## 2. Experimental Section

### 2.1. Setting and Participants

This study uses data obtained from the SQRP as most Swedish chronic pain clinics (>90%) refer data from the SQRP to assess patients and to develop interdisciplinary rehabilitation strategies. Typically, primary care physicians refer patients to these specialist clinics because the patients present with complex chronic pain. Patients complete the SQRP questionnaires on up to three occasions: during their first visit to the clinic (baseline); immediately after completing an interdisciplinary rehabilitation program; and at a 12-month follow-up. Not all patients participate in interdisciplinary rehabilitation programs and therefore do not report their situation on the second and third occasion. This study uses data from the first visit (baseline) only.

The study was conducted in accordance with the Helsinki Declaration and Good Clinical Practice and approved by the Ethical Review Board in Linköping (Dnr: 2015/108-31). All participants received written information about the study and gave their written consent.

### 2.2. Variables

The SQRP consists of several validated instruments that include questions about pain, psychological distress, perceived health, occupational status, and other demographic information. Since the SQRP’s inception in the 1990s, a few changes have been made; however, for the period from which data for this study were extracted (May 2007 to April 2016), no meaningful changes were made to the registry. This study uses measures similar to the recommendations by the IMMPACT group concerning outcomes in chronic pain trials [20] and more recent measures recommended for interdisciplinary pain therapy programs (i.e., VAPAIN) [21].

#### 2.2.1. Background Variables

The data set include age of the participant (years), gender (male or female), educational background (either college/university degree or not), and full-time employment status (yes or no). In addition, the dataset included how many times a patient visited a physician during the last year (four or more visits was considered as high health-care consumption).

#### 2.2.2. Pain Aspects

A Numeric Rating Scale (NRS) was used to rate average pain intensity for the last week, with a possible score from 0 to 10 where the highest number represents worst possible pain. This variable is denoted as *NRS-7 days*.

Pain intensity was determined using the Multidimensional Pain Inventory (MPI)—*MPI Pain Severity* (range 0–6). MPI is a 61-item questionnaire that measures the impact of chronic pain [22]. The current study uses the second version of the questionnaire in its Swedish translation [23]. 

A single question about whether the pain is persistent or recurrent (yes or no) was also included—*Persistent Pain*. In addition, the duration of chronic pain (years) was included—*Pain Duration*.

*Number of pain regions* (i.e., the degree of spreading of pain on the body) was obtained using 36 predefined anatomical areas (18 on the front and 18 on the back of the body), and the patients registered the areas where they experience pain: (1) head/face, (2) neck, (3) shoulder, (4) upper arm, (5) elbow, (6) forearm, (7) hand, (8) anterior aspect of chest, (9) lateral aspect of chest, (10) belly, (11) sexual organs, (12) upper back, (13) low back, (14) hip/gluteal area, (15) thigh, (16) knee, (17) shank, and (18) foot. The number of areas with pain (range: 1–36) were summed. For the current period in the SQRP, there was no option for indicating missing values for this variable, resulting in 4.5% of the participants indicating that they did not have any painful region of the body.

#### 2.2.3. Emotional Distress Variables

Symptoms of anxiety and depression were measured using the Hospital Anxiety and Depression Scale (*HADS*) [24], a 14-question instrument with seven questions addressing the anxiety subscale (denoted *HADS-A*) and seven the depression subscale (denoted *HADS-D*). Each item is scored between 0 and 3 with possible scores range from 0 to 21. Higher scores indicate more problems: a score of 7 or less on each subscale indicates a non-case, 8–10 indicates a possible case, and ≥11 indicates a definite case of anxiety/depression. The Swedish version of the instrument has sound psychometric properties [25]. The MPI subscale concerning emotional distress—*MPI Distress* (range 0–6)—was also used. Among these variables were a subscale of SF-36 (see below). Fatigue is a common complaint associated with EDS and HMS, so the *SF36-Vitality* subscale was included as a proxy for fatigue.

#### 2.2.4. Life Impact Variables

Within this area, three MPI variables were chosen: *MPI Interference, MPI Control,* and *MPI Activity*. *MPI Interference* measures interference of pain on one’s life, *MPI Control* measures the amount of perceived control of one’s life situation, and *MPI Activity*, measured using the General Activity Index of the MPI, measures one’s activity. These three subscales all range from 0 to 6.

#### 2.2.5. Health-Related Quality of Life (HRQoL) Variables

Widely used and with a validated Swedish translation, the Short Form Health Survey (*SF36*) is a multidimensional health concept that measures several facets of health and well-being [26]. The SF36 consists of eight dimensions, ranging from 0 to 100 that can be summed into a physical component summary (*SF36-PCS*) and a mental component summary (*SF36-MCS*). This study uses both the *SF36-PCS* and the *SF36-MCS*.

To assess HRQoL, the European Quality of Life Instrument (EQ-5D) was included [27]. The SQRP uses the EQ-5D-3L variant, which includes three possible levels on the five dimensions of the index. The present study included both the index (EQ-5D index) and the stand-alone question, where the participants are asked to estimate their current health on a thermometer-like vertical scale: 0 = worst possible state and 100 = perfect health (EQ-VAS).

### 2.3. Statistics

Unless otherwise mentioned, we used SPSS (version 24.0; IBM Corporation, Route 100 Somers, New York, USA) for descriptive analysis and group comparisons. We compared the different groups using a series of one-way ANOVAs. For categorical variables, dummy coding was performed followed by chi-square tests. As we did not assume equal variance between groups, Welch’s ANOVA, rather than Fischer’s, is reported. Effect sizes were calculated for the partial omega-squared statistic and the comparisons between EDS and HMS Cohen’s d are presented. For post hoc tests, we used the Games-Howell statistic. We employed the standard descriptions for ANOVAs of small (0.01–0.059), medium (0.06–0.139), and large (>0.14) effect sizes (ω_p_^2^) as suggested by Cohen [28]. Corresponding figures for the absolute Cohen’s d were insignificant for <0.20, small for 0.20–0.49, moderate for 0.50–0.79, large for 0.80–1.29, and very large for ≥1.3. For each step of the analysis, we generated a new Bonferroni corrected alpha-level to attain more conservative *p*-values in the context of multiple comparisons. The first part (Table 1) we used 0.0022 as the critical *p*-value. Table 2 contains no statistical testing. The main analysis was a series of ANOVAs combined with the chi-square tests that were set at a *p*-value of 0.05 divided by 24 (Table 3 and Table 4). For the Games-Howell post hoc tests, we used *p* = 0.0005 as the cut off.

Classical statistical methods (e.g., multiple regression) present a risk of downplaying the interrelationships among different factors and therefore reaching incorrect conclusions. Classical methods also assume variable independence when interpreting results and it can be risky to consider one variable at a time. If multicollinearity (i.e., high correlations) occurs among the X-variables, the regression coefficients become unstable and their interpretability breaks down. Hence, we used advanced multivariate data analysis (MVDA)—i.e., Principal Component Analysis (PCA)—for the multivariate correlation analyses to detect outliers and Orthogonal Partial Least Square Regressions-Discriminant Analysis (OPLS-DA) for the multivariate regressions of diagnoses. These analyses were performed using SIMCA-P+ (version 15.0; Sartorius Stedim Biotech, Umeå, Sweden) [29]. SIMCA-P+, in contrast to traditional statistical packages such as SPSS, uses the Nonlinear Iterative Partial Least Squares algorithm (NIPALS algorithm) when compensating for missing data for variables/scales (max 60%) and for subjects (max 50%). In the context of the obvious risks for multicollinearity problems, we refrained from using, for example, logistic regression. MVDA does not require normal distribution [30].

As outliers can markedly bias regressions, PCA was used to check for multivariate outliers. Outliers were identified using two methods: score plots in combination with Hotelling’s T^2^ and distance to model in X-space. R^2^ describes the goodness of fit—the fraction of sum of squares of all the variables explained by a principal component. Q^2^ describes the goodness of prediction—the fraction of the total variation of the variables that can be predicted by a principal component using cross validation methods.

OPLS-DA was used to explore which variables in the multivariate context differentiated EDS from the three other diagnoses; in these three regressions, EDS was denoted 1 and the other diagnoses were denoted 0 for the three comparisons. The variable influence on projection (VIP) indicates the relative relevance of each X-variable. VIP ≥ 1.0 was considered significant if the VIP value had a 95% jack-knife uncertainty confidence interval non-equal to zero. P(corr) was used to note the direction of the relationship (positive or negative). P(corr) depicts the loading of each variable scaled as a correlation coefficient with a standardized range from −1 to +1. For each regression, we report the R^2^, Q^2^, and the result (i.e., *p*-value) of a cross-validated analysis of variance (CV-ANOVA). In the present study, we required significant CV-ANOVA for a regression as a whole to be significant. A certain variable within a regression was considered significant when VIP > 1.0.

### 2.4. Analytical Approach

First, we wanted to determine whether there were any differences between patients with EDS and patients with HMS (step 1). Second, we wanted to determine whether there were potential differences between patients with either of these diagnoses and other common diagnostic groups of patients referred to specialist pain clinics (step 2). In the final step (step 3), we analyzed to what extent EDS multivariately differentiated from the three other diagnoses and which variables were important for these differences between diagnoses.

## 3. Results

### 3.1. Step 1

In the SQRP (*n* = 40,518), 427 patients were classified as with EDS and 420 patients were classified as with HMS. Since pain is a criterion for EDS hypermobility type we assumed that most of those patients belonged to this sub-type. As there was a potential for lack of precision in differential diagnosis between these related diagnoses, we looked for any systematic differences between these groups on the measures included in the study. No statistical differences (excluding one variable) between the two diagnoses were found (Table 1).

There was a significant difference on the SF36-PCS—i.e., patients with EDS scored somewhat lower (M = 25.2, SD = 8.13) than patients with HMS (M = 27.57, SD = 8.45) (t = (687.29) 3.50 *p* < 0.001, Cohen’s d = 0.27, interpreted as a small effect). In addition, no significant differences between the two groups for the categorical background variables (sex, educational level, and currently employed or studying), health-care consumption and persistent pain were revealed. Because the two groups exhibited similar results for the investigated variables, we clustered the two groups together in the further analysis (denoted EDS/HMS), resulting in a group of 795 patients (1.9% of the patients in SQRP) as some with EDS had a secondary diagnosis of HMS.

### 3.2. Step 2

To explore differences between patients with EDS/HMS and patients with other common conditions in the SQRP, we compared the EDS/HMS group with patients who had a main diagnosis of fibromyalgia, an approach used by previous research [16]. In addition, we wanted to compare EDS/HMS with more localized or regional musculoskeletal pain conditions in the spinal area (e.g., cervicobrachial syndrome, cervicalgia, low back pain, and thoracic spine pain); these conditions were labelled *spinal pain*. A more localized pain condition related to trauma—i.e., whiplash associated disorders (WAD)—was also included (Table 2). Therefore, we included four categories of patients (*n* = 14 203): spinal pain (*n* = 7183); fibromyalgia (*n* = 5791); WAD (*n* = 1229); and EDS/HMS (*n* = 795).

#### 3.2.1. Background Variables

Female gender was more common for EDS/HMS (93.6%) and fibromyalgia (95.1%) (Table 3). Age was associated with a large effect size; patients in the EDS/HMS group were younger (mean: 35.9 years) than the three other groups (39.4–46.6 years) (Table 4).

WAD had the highest proportion with high education (29.1%) and EDS/HMS (25.5%), whereas patients with fibromyalgia had the lowest proportion (19.8%) (Table 3). Working full time had the following distribution: WAD (50.4%), spinal pain (46.7%), EDS/HMS (46.2%), and fibromyalgia (38.7%) (Table 3). No group differences in health-care consumption were found.

#### 3.2.2. Pain Aspects

MPI Pain Severity, pain duration, and number of pain regions were associated with large to medium effect sizes (Table 4). Patients with fibromyalgia had the most problems with different aspects of pain such as intensity and severity. Although younger, the EDS/HMS patients had the longest pain duration (mean: 9.9 years) followed by the fibromyalgia group (8.7 years) (Table 4). On average, the EDS/HMS group had pain for five years longer than the WAD group and 3.8 years longer than the spinal pain group (both *p* < 0.0005). The EDS/HMS group had almost as much spread of pain on the body (i.e., number of pain regions) as the fibromyalgia group (mean: 19.49 vs. 22.28). The WAD group had 7.62 and the spinal pain group had 8.33 fewer sites than the EDS/HMS group (all *p* < 0.0005) (Table 4).

#### 3.2.3. Emotional Distress Variables

For the anxiety subscale of the HADS, there were significant group differences (Table 4). On a group level, all four groups scored between a possible case and a probable case (mean: 8.66–10.32). No group differences were larger than 1.66, indicating small clinical differences. For the depression subscale of HADS, the same kind of range between scores was found as for the anxiety subscale (mean: 8.14–9.52), with the largest group difference being only 1.38. MPI distress showed a similar pattern as the two subscales of HADS. The SF36-Vitality scale, a proxy for fatigue, was associated with a large effect size. All post hoc tests were significant (Table 4). In short, EDS/HMS (mean: 19.92) and fibromyalgia (17.26) had lower vitality than the spinal pain group (27.06) (both *p* < 0.0005) and WAD was intermediary.

#### 3.2.4. Life Impact Variables

Fibromyalgia had the highest scores on MPI Interference (mean: 4.61) and without significant differences between EDS/HMS and the two other groups (4.32–4.43). The worst situation also existed for fibromyalgia (mean: 2.40) with respect to MPI Control. EDS/HMS and WAD were intermediary (both 2.59) and the best situation was found for the spinal pain group (2.81). MPI Activity only showed trivial differences across the four groups.

#### 3.2.5. Health-Related Quality of Life

SF-36 PCS was associated with a large effect size (Table 4). There was no significant difference between EDS/HMS and fibromyalgia. In contrast, both the spinal pain group (29.85) and the WAD group (30.87) rated their physical health as better than the EDS/HMS group (both *p* < 0.0005). The three other variables within this area—SF-36 MCS, EQ-5D index, and EQ-VAS—showed significant differences, but the effect sizes were small.

### 3.3. Step 3 Multivariate Regressions of Diagnoses

In the next step, we used the variables under investigation to multivariately investigate (i.e., all variables taken together considering their complex interrelationships) whether there were significant differences in the clinical presentations between EDS/HMS and the three other diagnoses (Table 5). Number of pain regions was excluded in these regressions since it is part of diagnostic criteria for fibromyalgia and EDS/HMS. The three regressions were all composed by one predictive component and one orthogonal component. Although the three regressions were highly significant, they explained only 9–29% of the group belonging (i.e., the diagnoses), i.e., no prominent differences in clinical presentations according to the used PROMs existed.

The variables that significantly differentiated EDS/HMS from fibromyalgia were age, two pain intensity variables (NRS-7days and MPI Pain Severity), depressive symptoms, and SF36-MCS (Table 5). Hence, fibromyalgia had on average higher age and somewhat more severe pain intensity and psychological strain than EDS/HMS. However, the explained variation was less than 10% (R^2^ and Q^2^ were both 0.09). 

In the multivariate context, variables that significantly contributed to differentiate spinal pain from EDS were female gender, age, pain duration, SF36-PCS, and SF36-Vitality (Table 5). Hence, there were relatively more women with EDS/HMS, lower age in EDS/HMS, longer pain duration in EDS/HMS, and lower vitality and sf36-PCS in EDS/HMS. In addition, this regression explained a low part of belonging to spinal pain or EDS/HMS diagnoses since both R^2^ and Q^2^ were low (0.13).

The strongest regression was obtained when multivariately differentiating between WAD and EDS/HMS (Table 5). Pain duration, female gender, sf36-PCS, and age were significant regressors. Hence, compared to WAD, EDS/HMS was associated with longer pain duration, higher female proportion, lower sf36-PCS, and lower age.

## 4. Discussion

This study reveals several overlapping as well as condition-specific impacts of chronic pain for patients with EDS/HMS, fibromyalgia, spinal pain, and WAD. Six variables showed significant differences with large effects sizes—age, pain duration, number of pain regions, MPI Pain Severity, SF36 -Vitality, and SF36-PCS) (Table 4). Compared to other diagnostic groups, patients with EDS/HMS were younger, more often female, and had suffered from pain for the longest time. The impact of chronic pain on daily life was of similar magnitude for EDS/HMS and fibromyalgia in several aspects (e.g., fatigue and physical health); however, these two diagnostic groups did not totally overlap as the analysis in step 3 revealed that they differed in age, pain intensity, and somewhat on depressive symptoms. In other areas such as Pain Severity/Intensity, Anxiety, and Depressive Symptoms, EDS/HMS was similar to spinal pain and WAD. Multivariate analyses of the self-reports did not identify a very distinct clinical presentation for EDS/HMS.

### 4.1. EDS vs. HMS—Step 1

We did not find any important significant differences between patients with EDS and patients with HMS diagnoses (Table 1). HMS and EDS hypermobility type are considered two different connective tissue disorders with regard to possible genetic origin, diagnostics criteria, and severity [34,35]. However, because these disorders have very similar clinical features and consequences, they are usually considered the same disorder from clinical and treatment perspectives [12,36,37,38].

### 4.2. Step 2

#### 4.2.1. Age, Pain Duration, and Gender

The patients with EDS/HMS were younger and had the longest pain duration. This result is expected as EDS/HMS is a genetic disorder that debuts in childhood, and its consequences, including pain, increase as patients age [39]. Women were overrepresented in EDS/HMS and fibromyalgia, a finding confirmed in other studies [2,40,41]. This gender difference in EDS/HMS might be the result of social, genetic, and biological factors [41].

#### 4.2.2. Number of Pain Regions

Compared with patients who had spinal pain or WAD patients, fibromyalgia and EDS/HMS patients reported significantly more painful sites. This is expected since both criteria for fibromyalgia and EDS/HMS require spreading of pain. Widespread pain, a feature of several chronic pain disorders [42], negatively impacts mood, cognition, fatigue, and work status [43,44]. Chronic widespread pain has been associated with alterations in the brain, neuroinflammation, central sensitization, systemic low grade inflammation, and nociceptor and muscle alterations [45,46,47,48,49,50,51,52,53]. Sensitization of the nociceptive system has been presented as a possible cause both for fibromyalgia-related [54] and hypermobile EDS/HMS-related pain [55].

#### 4.2.3. Pain Severity and Intensity

The MPI scale Pain Severity, which measures perceived severity and intensity of pain, was rather high for all diagnostic groups, but the fibromyalgia group scored significantly higher. A similar pattern was found for the other pain intensity variable NRS-7 days, confirming results from previous research [16]. Whether the development of chronic pain is a late consequence of joint hypermobility or a hypermobility-related co-morbidity is still a matter of debate and further research is needed to clarify this point [9].

#### 4.2.4. Vitality/Fatigue

Fatigue is one of the main explanations for disability among EDS patients [7]. Although all diagnostic groups reported low levels of vitality (the proxy for fatigue) (i.e., 17.26–27.06; Swedish population norm: 68.8) [26], fibromyalgia and EDS/HMS experienced more fatigue (lower vitality) compared to the two other diagnostics groups. However, the Swedish National EDS Association reports higher SF-36 Vitality scores (mean 30.2; CI 27.8 to 33.0) [3] for their members than what we report. The lower vitality scores in the present patients were likely due to selection of the most severe cases since they were referred to specialist clinics. Moreover, the Swedish National EDS Association included different sub-types of the EDS, while in our study the sub-types could not be determined.

Although the fatigue reported by EDS/HMS patients has usually been related to chronic pain and its consequences such as muscle weakness [55,56] and kinesiophobia [57], no evidence suggests that the fatigue is specifically related to EDS/HMS. Rather, fatigue is considered to be a result of a broader context of several somatic and psychological issues [58,59]. As in our study, several other studies have found that EDS/HMS and fibromyalgia patients, compared to the general Swedish population, experience lower HRQoL (i.e., SF36-PCS and SF36-MSC) often associated with chronic fatigue [3,60,61,62].

#### 4.2.5. Emotional Distress

There is an increasing amount of evidence pointing toward a high prevalence of psychiatric conditions among individuals with ED/HMS in general and anxiety disorders in particular [61]. However, there are some controversies regarding the psychopathology associated with EDS/HMS (i.e., whether it is a part of EDS/HMS itself or it is the manifestation of associated conditions such as chronic pain, sleep disorders, or fatigue). However, we did not find any significant association regarding emotional distress between EDS/HMS and spinal pain or WAD patients.

### 4.3. Multivariate Considerations—Step 3

The results from step 3 highlighted several differences and similarities between the groups. The overall largest differences existed between WAD and EDS/HMS, with and R^2^-value of 0.29. This finding was perhaps not that surprising considering the different pathways to a pathological status between these diagnoses. The clear differences only existed for variables such as duration of the condition, age, sex, and physical quality of life. Therefore, although these diagnoses might be considered to be very different, the patients who seek specialized health-care for these diagnoses in many aspects are quite similar—at least when considering how they self-rate their own health status.

### 4.4. Strengths and Limitations

This study’s main strength is its large cohort of patients: 795 patients with either EDS or HMS. To the best of our knowledge, this study is one of the largest studies of these diagnoses. In the SQRP, the proportion of EDS/HMS patients was 1.9%, somewhat smaller than the estimated proportion of 3% for the general population [2]. This difference could be attributed to the fact that the patients were referred to specialist care. Thus, our results were reasonably representative for EDS/HMS patients with complex chronic pain conditions. One weakness of this study concerns diagnostics. The patients were allocated to diagnostic groups according to the relevant international classification of disease (ICD-10) codes reported in the SQRP. Patients with EDS were defined according to ICD-10 code Q79.6 and hypermobility syndrome according to M35.7. However, this study did not validate these diagnoses since we lacked information on Beighton score and other specific clinical data. Moreover, the ICD-10 code for EDS does not distinguish between types of EDS such as classical, hypermobile, or vascular. Furthermore, in some situations it may be difficult to clinically distinguish between fibromyalgia and EDS/HMS.

### 4.5. Clinical Relevance

Our findings indicate that EDS/HMS patients referred to pain rehabilitation specialist clinics have complex pain conditions affecting broad aspects of their life, the reported long duration of pain might have contributed to the generalized pain condition similar to fibromyalgia. Persons with EDS/HMS might have an early onset of pain. These results point at the importance of clinical awareness e.g., in primary care that young persons with EDS/HMS are at risk of developing a chronic pain condition.

The results also indicate that assessment needs to address not only medical diagnosis, but also pain, function, activity, and participation. Interventions that consider all these aspects to prevent or treat negative long-term consequences might be needed. Relatively complex interventions (interdisciplinary multimodal rehabilitation programs) may be necessary to improve the patients’ global situation [35,63,64].

## 5. Conclusions

Our results point out that EDS/HMS is associated with a consistent burden of disease like that of fibromyalgia and partially worse than spinal pain and WAD. According to the multivariate analyses of self-reports, there is not a very distinct clinical presentation for EDS/HMS. Moreover, broad impact of chronic pain on daily life was noted for the group of patients with EDS/HMS. Health-care clinicians must be aware that complex rehabilitation interventions might be necessary to improve their patient’s global situation.

## Figures and Tables

**Table 1 jcm-09-02143-t001:** Comparison between those coded with Ehlers–Danlos syndrome (EDS) and those with Hypermobility syndrome (HMS). For the categorial variables (upper part of the table) are given per centages and for the non-categorical variables (lower part of the table) mean and standard deviation (SD). Furthest to the right is presented the results of the statistical comparisons, Chi-square tests and Welch’s *t* test, respectively. For the non-categorical variables are presented with corresponding effect sizes (Cohen’s d).

Group	EDS		HMS		Statistics			
Variables					Chi^2^ ^‡^			
Women	94%		93%		0.7
Higher education	24%		26%		0.44
Working/studying full time	42%		47%		4.65
High health-care consumption	65%		65%		0.05
Persistent pain	86%		88%		9.21
	Mean	SD	Mean	SD	Welch’s t	df	*p*	Cohen’s d
Age (years)	36.2	10.56	35.5	11.09	1.17	741.69	0.24	0.09
NRS 7 days	6.71	1.57	7.00	1.61	0.95	509.87	0.34	0.08
MPI Pain Severity	4.37	0.86	4.48	0.87	0.69	717.24	0.49	0.05
Pain duration (years)	14.00	11.08	11.25	10.32	2.55	433.79	0.01	0.24
Number of pain regions	20.12	8.61	18.77	8.38	−0.46	730.30	0.65	−0.03
HADS-A	8.8	4.83	9.42	4.99	1.82	743.53	0.07	0.13
HADS-D	8.29	4.50	8.14	4.38	−1.37	719.99	0.17	−0.10
MPI Distress	3.43	1.26	3.56	1.30	−0.69	713.08	0.49	−0.05
SF36 Vitality	19.36	16.73	20.55	18.63	0.13	685.26	0.90	0.01
MPI Interference	4.49	0.99	4.36	1.07	−1.44	711.67	0.15	−0.11
MPI Control	2.56	1.05	2.62	1.12	1.71	705.89	0.09	0.13
MPI Activity	2.29	0.78	2.33	0.85	−1.13	713.74	0.26	−0.08
SF36 PCS	25.2	8.13	27.57	8.45	−3.50	687.29	< 0.001 *	−0.27
SF36 MCS	37.13	12.96	36.2	13.41	−0.87	696.12	0.38	−0.07
EQ-5D Index	0.22	0.30	0.24	0.30	−0.23	708.70	0.82	−0.02
EQ-VAS	40.28	18.35	40.49	19.86	−0.69	690.88	0.49	−0.05

Note. ^‡^ = No significant value for the categorial variables at the 0.0022 level. * = significant at the 0.0022 level, EDS: Ehlers—Danlos syndrome; HMS: hypermobility syndromes. NRS—Numeric Rating Scale; MPI—Multidimensional Pain Inventory; HADS—Hospital Anxiety and Depression Scale; SF-36—Short Form Health Survey 36; EQ-5D—European Quality of Life Instrument.

**Table 2 jcm-09-02143-t002:** ICD-10-SE codes, number of patients for each code and the designation groups in the present study for the reference groups of chronic pain patients without Ehlers–Danlos syndrome or Hypermobility syndrome (EDS/HMS).

Name	Code	*n*	Designation
Cervicobrachial Syndrome	M53.1	2230	Spinal pain
Cervicalgia	M54.2	1402	Spinal pain
Low Back Pain	M54.5	3061	Spinal pain
Pain in Thoracic Spine	M54.6	490	Spinal pain
Fibromyalgia	M79.7	5791	Fibromyalgia
Whiplash	S13.4	532	WAD
Sequelae, Whiplash	T91.8	697	WAD

WAD—Whiplash Associated Disorders.

**Table 3 jcm-09-02143-t003:** Percentage (%) for background variables and persistent pain (i.e., the categorical variables) in the four groups of patients. Furthest to the right is reported the statistical comparison (Chi-square) between the four groups of patients.

Variable	EDS/HMS	WAD	Spinal Pain	Fibromyalgia	Chi^2^
Women	93.6%	64%	63.2%	95.1%	2058.14 *
Higher Education	25.5%	29.1%	22.7%	19.8%	56.4 *
Working/Studying Full Time	46.2%	50.4%	46.7%	38.7%	99.19 *
High Health-Care Consumption	69.0%	68.8%	69.6%	69.4%	0.37
Persistent Pain	88.9%	86.8%	84.5%	91.3%	129.28 *

Note. * = significant at the 0.0022 level. For the years in question, 27.85% of Swedish women had a college/university degree [31]. The proportion of women in active in working life was 73.27% from 2008 to 2017 [32]. For 2016, 68.6% was working full time [33]. Please note that there are differences in definitions between average population numbers and data from the SQRP.

**Table 4 jcm-09-02143-t004:** Mean values (and one standard deviation) for the Patient Reported Outcome Measures (continuous variables) in the four groups of patients. The results of the one-way ANOVAs comparing the four groups (i.e., DF, Welch’s F, and ω_p_^2^) are to the right.

Group	EDS/HMS	WAD	Spinal Pain	Fibromyalgia	DF	Welch’s F	ω_p_^2^
Age (years)	35.87 (10.81) ^b,c,d^	39.4 (10.59) ^a,c,d^	46.63 (11.13) ^a,b^	43.57 (10.34) ^a,b^	3, 2611.73	174.87 *	0.17
NRS-7 days	6.85 (1.6) ^d^	6.93 (1.76) ^d^	6.87 (1.79) ^d^	7.46 (1.59) ^a,b,c^	3, 2529.83	140.34 *	0.14
MPI Pain Severity	4.42 (0.87) ^d^	4.43 (0.94) ^d^	4.38 (0.96) ^d^	4.72 (0.83) ^a,b,c^	3, 2522.07	155.11 *	0.16
Pain duration (years)	9.9 (10.08) ^b,c,d^	4.84 (6.97) ^a,c,d^	6.15 (7.64) ^a,b,d^	8.71 (8.61) ^b,c^	3, 2203.97	218.79 *	0.23
Number of pain regions	19.49 (8.52) ^b,c,d^	11.88 (6.69) ^a,d^	11.17 (6.95) ^a,d^	22.28 (8.09) ^a,b,c^	3, 2589	2408.24 *	0.74
HADS-A	9.09 (4.91) ^d^	9.72 (4.89) ^c^	8.66 (4.89) ^b,d^	10.32 (4.98) ^a,c^	3, 2529	116.94 *	0.12
HADS-D	8.22 (4.44) ^d^	8.93 (4.73) ^c^	8.14 (4.63) ^b,d^	9.52 (4.61) ^a,c^	3, 2528,25	95.37 *	0.1
MPI Distress	3.49 (1.28) ^d^	3.68 (1.27) ^c^	3.37 (1.35) ^b,d^	3.75 (1.27) ^a,c^	3, 2534.88	90.58 *	0.1
SF36-Vitality	19.92 (17.64) ^b,c^	24.37 (18.84) ^a,c,d^	27.06 (19.69) ^a,b,d^	17.26 (16.12) ^b,c^	3, 2476.1	310.51 *	0.27
MPI Interference	4.43 (1.03) ^d^	4.41 (1.1) ^d^	4.32 (1.08) ^d^	4.61 (0.95) ^a,b,c^	3, 2484.89	81.96 *	0.09
MPI Control	2.59 (1.09) ^c,d^	2.59 (1.13) ^c,d^	2.81 (1.17) ^a,c,d^	2.40 (1.14) ^a,b,c^	3, 2538.85	130.99 *	0.13
MPI Activity	2.31 (0.82)	2.3 (0.93) ^c^	2.42 (0.89) ^b,d^	2.34 (0.87) ^c^	3, 2532.75	12.75 *	0.01
SF36-PCS	26.32 (8.36) ^b,c^	30.87 (7.93) ^a,c,d^	29.85 (7.98) ^a,b,d^	26.43 (7.37) ^b,c^	3, 2364.4	240.88 *	0.23
SF36-MCS	36.69 (13.17) ^b,d^	33.89 (13.1) ^a,c^	36.87 (13.37) ^b,d^	33.48 (12.48) ^a,c^	3, 2384.43	72.18 *	0.08
EQ-5D Index	0.23 (0.3)	0.25 (0.32) ^d^	0.27 (0.31) ^d^	0.21 (0.31) ^b,c^	3, 2436.37	36.44 *	0.04
EQ-VAS	40.38 (19.06) ^d^	40.76 (20.34) ^d^	42.61 (20.31) ^d^	37.16 (19.26) ^a,b,c^	3, 2440.44	73.51 *	0.08

Note. * *p* < 0.001. Games–Howell post hoc test indicated with letter if *p* <.0005. Effect size is reported in the ω_p_^2^ form. ^a^ significant compared to EDS/HMS. ^b^ significant compared to WAD. ^c^ significant compared to spinal pain. ^d^ significant compared to fibromyalgia. DF = Degrees of freedom.

**Table 5 jcm-09-02143-t005:** OPLS-DA regressions differentiating EDS/HMS from FM, spinal pain, and WAD, respectively. Variables in bold type are significant.

FM vs. EDS/HMS	VIP	*p* (corr)	Spinal vs. EDS/HMS	VIP	*p* (corr)	WAD vs. EDS/HMS	VIP	*p* (corr)
Age	2.97	−0.82	Women	2.62	0.63	Pain duration	2.65	0.67
NRS-7days	1.61	−0.44	Age	2.19	−0.52	Women	2.55	0.66
MPI Pain Severity	1.44	−0.39	Pain duration	1.64	0.38	SF36-PCS	1.57	−0.40
HADS-D	1.10	−0.30	SF36-PCS	1.49	−0.35	Age	1.13	−0.30
SF36-MCS	1.03	0.28	SF36-Vitality	1.30	−0.31	SF36-Vitality	0.99	−0.26
MPI Distress	0.87	−0.24	Persistent pain	0.58	0.14	SF36-MCS	0.71	0.18
HADS-A	0.85	−0.23	MPI Control	0.54	−0.13	HADS-D	0.49	−0.13
MPI Interference	0.82	−0.22	MPI Interference	0.45	0.11	MPI Distress	0.46	−0.12
Working/studying full time	0.66	0.18	EQ-VAS	0.45	−0.11	Working/studying full time	0.45	−0.12
EQ-VAS	0.65	0.18	EQ-5D Index	0.39	−0.09	HADS-A	0.45	−0.12
MPI Control	0.57	0.15	MPI Pain Severity	0.37	0.09	NRS-7days	0.30	−0.07
Persistent pain	0.46	−0.13	Higher education	0.34	0.08	MPI Interference	0.24	0.06
SF36-Vitality	0.44	0.12	SF36-MCS	0.31	0.07	Higher education	0.23	−0.06
Higher education	0.42	0.12	MPI Activity	0.29	−0.07	Persistent pain	0.23	0.06
EQ-5D Index	0.41	0.11	MPI Distress	0.16	0.04	EQ-5D Index	0.22	−0.06
Women	0.24	−0.07	NRS-7days	0.16	0.04	EQ-VAS	0.18	−0.04
MPI Activity	0.23	−0.07	High health-care consumption	0.16	−0.04	MPI Activity	0.16	0.04
High health-care consumption	0.10	−0.03	HADS-D	0.09	−0.02	High health-care consumption	0.09	−0.02
Pain duration	0.06	0.01	Working/studying full time	0.07	−0.02	MPI Pain Severity	0.07	−0.02
SF36-PCS	0.02	−0.01	HADS-A	0.05	0.01	MPI Control	0.01	0.00
R^2^	0.09		R^2^	0.13		R^2^	0.29	
Q^2^	0.09		Q^2^	0.13		Q^2^	0.29	
*p*-value	<0.001		*p*-value	<0.001		*p*-value	<0.001	
*n*	6245		*n*	7686		*n*	1972	

Note. VIP (VIP > 1.0 is significant) and *p* (corr) are reported for each regressor i.e., the loading of each variable scaled as a correlation coefficient and thus standardizing the range from −1 to +1. EDS is denoted 1 and the other diagnoses in each regression denoted as 0. The sign of *p* (corr) indicates the direction of the correlation with the dependent variable (+ = positive correlation; − = negative correlation). The four bottom rows of each regression report R^2^, Q^2^, *p*-value of the CV-ANOVA, and number of patients. FM = Fibromyalgia. VIP = Variable Influence on Projection.

## Data Availability

The datasets generated and/or analyzed in this study are not publicly available as the Ethical Review Board has not approved the public availability of these data.

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
