# Peer review of "Ehlers–Danlos Syndrome and Hypermobility Syndrome Compared with Other Common Chronic Pain Diagnoses—A Study from the Swedish Quality Registry for Pain Rehabilitation"

_jcm, 2020, doi:10.3390/jcm9072143_

Round 1

Reviewer 1 Report

The manuscript “Ehlers-Danlos Syndrome and Hypermobility syndrome compared with other common chronic pain diagnoses – A study from the Swedish Quality Registry for Pain Rehabilitation” by Molander et al. attempts to characterize chronic pain and its associated effects in EDS/HMS and to compare it to other chronic and local pain conditions based on Patient Reported Outcome Measures (PROMs) in large cohort of patients using data from the Swedish Quality Registry for Pain Rehabilitation (SQRP). Data was collected from the years 2007-2016.

The field of pain research in hypermobile EDS is significantly lacking despite the relatively high prevalence of the condition. This is an interesting manuscript and I recommend accepting it for publication with acknowledgement of the deficiencies in its presented work. The manuscript’s strength is in the large numbers and in the comparison of the pain in EDS/HMS to other both generalized and localized chronic pain conditions.    

The main weakness of this manuscript is the absence of confirmed diagnosis for the different EDS sub types (which have significantly different clinical presentation). The data were collected prior to the implementations of the 2017 diagnostic criteria and the EDS/HMS – groups m/p contain overlapping conditions. Although the introduction mentions in general the diagnostic criteria of hEDS, it should give the reader more detailed background of the different types of EDS and of the diagnostic criteria that were accepted for hEDS prior to 2017. This will help to put this work in the appropriate context (can be added as a supplemental data). Please consider to omit the word “possible” from the sentence in line 381 “One possible weakness of this study concern”.

The authors do not mention the high prevalence of co-morbidities frequently seen in hEDS patients including various autoimmune conditions. How many of the presented EDS / HMS patients in this cohort had an additional clinical diagnosis other than EDS?

“Number of pain regions (i.e., the degree of spreading of pain on the body) was obtained using 36 predefined anatomical areas” – Can authors give any information regarding differences in the areas of pain selected in the EDS groups vs the other groups?

Line 392 “The early onset of chronic pain might coincide with starting a family and a career, entailing severe strain on those affected”. This statement suggests that EDS patients are more prone to psychosocial stressors than others. Your data does not suggest this (section 4.2.5.). Also, authors show no other data to support this statement. Do these patients start family and career earlier than patients with fibromyalgia or the other tested conditions?

There are no healthy population controls in this study. Yet, it could be interesting to know what the employment rates of women and the rate of women with high education in the general population in Sweden between the years were 2007-2016. This will give some information regarding the debilitating effect of the tested conditions.

Author Response

Thank you for the positive attitude towards our manuscript and the constructive comments!

Please se below for our point-to-point reply:

1.      The main weakness of this manuscript is the absence of confirmed diagnosis for the different EDS sub types (which have significantly different clinical presentation). The data were collected prior to the implementations of the 2017 diagnostic criteria and the EDS/HMS – groups m/p contain overlapping conditions. Although the introduction mentions in general the diagnostic criteria of hEDS, it should give the reader more detailed background of the different types of EDS and of the diagnostic criteria that were accepted for hEDS prior to 2017. This will help to put this work in the appropriate context (can be added as a supplemental data).

We added two sentences in the introduction about the older Villefranche nosology and a reference for a new supplement (Supplement 1) with more information about the diagnostic criteria prior to 2017 on line 51-53.

We also added information on line 208-209 about pain often being a feature of hypermobile EDS and that we thus assumed that most of patients belonged to this subtype.

2.       Please consider to omit the word “possible” from the sentence in line 381 “One possible weakness of this study concern”.

We have now removed the word "possible" as suggested from line 405 (for some reason line numbers do not appear to be matching).

3.      The authors do not mention the high prevalence of co-morbidities frequently seen in hEDS patients including various autoimmune conditions. How many of the presented EDS / HMS patients in this cohort had an additional clinical diagnosis other than EDS?

Regarding other diagnosis and co-morbidities: We concur in that this is a highly relevant question. Unfortunately, the SQRP only mandates that one ICD-code is submitted for each patient although there is room for up to five codes per patient. When examining the coding of comorbidities, there were only sporadic use of these extra spaces and we thus decided not to use them at all (other than searching for patients with a connective tissue disorders)

4.      “Number of pain regions (i.e., the degree of spreading of pain on the body) was obtained using 36 predefined anatomical areas” – Can authors give any information regarding differences in the areas of pain selected in the EDS groups vs the other groups?

Regarding painful areas of the body: We omitted comparisons here as the manuscript already contained a lot of results. One future possibility would be to publish this data as a separate publication, drawing inspiration from Rombaut et al (2015) who published some results from similar drawings.

5.      Line 392 “The early onset of chronic pain might coincide with starting a family and a career, entailing severe strain on those affected”. This statement suggests that EDS patients are more prone to psychosocial stressors than others. Your data does not suggest this (section 4.2.5.). Also, authors show no other data to support this statement. Do these patients start family and career earlier than patients with fibromyalgia or the other tested conditions?

We realized we failed in clearly communicating what we meant regarding family life. What we wished to convey was that our patients had severe pain intensity and pain (quite similar to FMS), while being about 10 years younger. This meant that they must face the challenge of starting a family and/or career whilst suffering from severe pain, while the other diagnostic groups were older when they came to specialist level health care.

We have now re-written section 4.5 (line 415-423).

6.      There are no healthy population controls in this study. Yet, it could be interesting to know what the employment rates of women and the rate of women with high education in the general population in Sweden between the years were 2007-2016. This will give some information regarding the debilitating effect of the tested conditions.

The average employment/studying rate for women for the current years was 73.27%

Average higher education rate for women was 27.85%. This information has now been added to Table 3 but should be interpreted with some caution as the phrasing in the SQRP is not exactly the same as from the official population-wide data.

Reviewer 2 Report

In this study, the authors interpreted patient reported data from the Swedish Quality Register for Chronic Pain for patients with ‘Ehlers-Danlos syndrome (EDS)’ and hypermobility syndrome (HMS) and compared this patient group with fibromyalgia patients and 2 patient groups with localized pain conditions: spinal pain and whiplash associated disease. Several variables were studied and revealed that the pain and other parameters associated with ‘EDS’/HMS are between fibromyalgia and localized pain conditions.

Please find my comments and suggestions below:

  • Often, references that are used refer to findings in a hypermobile EDS (hEDS)/hypermobility syndrome cohort but the way the text is written it appears to apply to the general EDS population (e.g. ref 13, 15, 17, 18, 52; …). This should be carefully checked throughout the manuscript and adapted to hEDS when necessary, since pain, fatigue, etc. are mainly studied and described for the hypermobile EDS/(joint) hypermobility syndrome patients and only limited data is reported for other EDS subtypes.
  • Line 43: The prevalence of EDS is stated to be 1%, but this is not correct. The authors use a reference that is specific for hypermobile EDS/hypermobility syndrome and not EDS in general. This needs to be adapted.

  • Is the selected time frame of data collection (2007-2016) due to the alterations in the registry that occurred over time as the authors describe? Or is there another reason for this? Given the selected time frame, the EDS diagnoses were most likely made based on the clinical criteria defined in the Villefranche nosology?

  • In the analysis is step 1 comparing the ‘EDS’ group with the HMS group, one value was significantly different: SF36-PCF, however in Table 1, the significant P-value of <0.001 is indicated for SF36-MCF.

  • Line 220: the number of patients with ‘spinal pain’ should probably be higher, based on Table 2? I think ‘pain in thoracic spine’ was not included in the counting (also in the total counting), assuming that this group of patients (n = 490) is included in the analyses?

  • Is there any data available about medication use in the populations?

  • For the pain duration (in years), the standard deviation is larger than the mean in most groups, this means that, despite significance, the variation for this parameter is large?

  • Table 5 indicates the difference of the different pain conditions with EDS, but should probably be the combined EDS/HMS group?

  • Line 295-296: This sentence is not entirely clear, and should probably be rephrased: “In addition, this regression explained a low part of belonging to spinal pain or EDS/HMS diagnoses.”

  • Line 316: In the discussion, the authors refer for the first time to their cohort as EDS hypermobility type. This needs to be addressed a bit more, given that no information was available on the EDS subtype before.

  • Do the authors think that an additional reason why the vitality/fatigue scores from their study are lower compared to the reported scores from the Swedish National EDS Association can also be due to the fact that different subtypes were included in the previous study, whereas in this study the subtypes could not be determined?

Author Response

Thank you for the very constructive comments!

Please see below for our point-to-point reply:

1.      Often, references that are used refer to findings in a hypermobile EDS (hEDS)/hypermobility syndrome cohort but the way the text is written it appears to apply to the general EDS population (e.g. ref 13, 15, 17, 18, 52; …). This should be carefully checked throughout the manuscript and adapted to hEDS when necessary, since pain, fatigue, etc. are mainly studied and described for the hypermobile EDS/(joint) hypermobility syndrome patients and only limited data is reported for other EDS subtypes.

This has been helpful, and we now have added "hypermobile" before typing EDS when appropriate: (line 56, line 57, line 60 and 62, 357)

2.      Line 43: The prevalence of EDS is stated to be 1%, but this is not correct. The authors use a reference that is specific for hypermobile EDS/hypermobility syndrome and not EDS in general. This needs to be adapted.

Thank you for noticing this. The sentence has now been rearranged to make it clear that the percentage alludes to hypermobile EDS (line 43-44).

3.      Is the selected time frame of data collection (2007-2016) due to the alterations in the registry that occurred over time as the authors describe? Or is there another reason for this? Given the selected time frame, the EDS diagnoses were most likely made based on the clinical criteria defined in the Villefranche nosology?

The registry underwent a large change at the end of the timeframe for this dataset, and that is the reason for the years we used in this study. Regarding diagnostic criteria, Reviewer 1 had the same question, and this has been clarified in a new supplement.

4.      In the analysis is step 1 comparing the ‘EDS’ group with the HMS group, one value was significantly different: SF36-PCF, however in Table 1, the significant P-value of <0.001 is indicated for SF36-MCF.

Thank you so much for noticing this! The PCF and MCF values had traded places when reformatting the table. This also made us aware of minimal difference between the correctly reported value in the table and an earlier draft version in the text (where the effect size was wrongly reported as 0.29 when the correct value is 0.27). These changes are on line 224-225.

5.      Line 220: the number of patients with ‘spinal pain’ should probably be higher, based on Table 2? I think ‘pain in thoracic spine’ was not included in the counting (also in the total counting), assuming that this group of patients (n = 490) is included in the analyses?

Another embarrassing omission. Thank you for noticing! In all analysis, the “pain in thoracic spine”-group was included so the correct value is now reported as 7183 (line 239).

6.      Is there any data available about medication use in the populations?

Unfortunately, the SQRP does not collect data about medication use.

7.      For the pain duration (in years), the standard deviation is larger than the mean in most groups, this means that, despite significance, the variation for this parameter is large?

The reason for this is that this variable had a long “tail”, positive skew, with some individuals having had pain for several decades while most only had had it for a couple of years. One can consider if the median had been a better choice here.

8.      Table 5 indicates the difference of the different pain conditions with EDS, but should probably be the combined EDS/HMS group?

This has now been clarified with the use of “EDS/HMS” instead of just EDS (Line 310).

9.      Line 295-296: This sentence is not entirely clear, and should probably be rephrased: “In addition, this regression explained a low part of belonging to spinal pain or EDS/HMS diagnoses.”

Added “since both R2 and Q2 were low (0.13).” at the end of the sentence to clarify, line 319.

10.   Line 316: In the discussion, the authors refer for the first time to their cohort as EDS hypermobility type. This needs to be addressed a bit more, given that no information was available on the EDS subtype before.

We added a sentence about possible EDS subtype in our cohort in the beginning of the result section, line 208-209.

11.   Do the authors think that an additional reason why the vitality/fatigue scores from their study are lower compared to the reported scores from the Swedish National EDS Association can also be due to the fact that different subtypes were included in the previous study, whereas in this study the subtypes could not be determined?

That is likely the case. In the revised version we have added a new sentence at line 374-376 expanding on this.

Reviewer 3 Report

Dear authors

I thank the authors for this relevant study, especially because of the clinical importancy. The chosen method seems very appropriate.

I have some comments, especially for the discussion, clinical relevance and conclusions.

The conclusions and clinical relevance are overlapping and should be more detailed based on the results. More focus should be on the types of interventions, based on the results.

it would be important to give the clinicians more structure, both for the choices in assessment tests related to the results  as for the types of intervention for the relevant subgroups.

without these addings the relevant work will not lead to implementation in the field of clinicians  

Author Response

Thank you for the positive attitude towards our manuscript and the constructive comments!

Please see below for our response:

1.      The conclusions and clinical relevance are overlapping and should be more detailed based on the results. More focus should be on the types of interventions, based on the results. it would be important to give the clinicians more structure, both for the choices in assessment tests related to the results as for the types of intervention for the relevant subgroups. without these addings the relevant work will not lead to implementation in the field of clinicians 

Thank you for this suggestion. As a result, we have extensively rewritten the “Clinical Relevance” section, line 415 to 423. This will hopefully address the needed improvements, while still not overstating the type of conclusions that we can draw from cross-sectional data.